# ORACLEAGENT: A MULTIMODAL REASONING AGENT FOR ORACLE BONE SCRIPT RESEARCH

## ABSTRACT

As one of the earliest writing systems, Oracle Bone Script (OBS) preserves the cultural and intellectual heritage of ancient civilizations. However, current OBS research faces two major challenges: **(1)** the interpretation of OBS involves a complex workflow comprising multiple serial and parallel sub-tasks, and **(2)** the efficiency of OBS information organization and retrieval remains a critical bottleneck, as scholars often spend substantial effort searching for, compiling, and managing relevant resources. To address these challenges, we present **OracleAgent**, the first agent system designed for the structured management and retrieval of OBS-related information. OracleAgent seamlessly integrates multiple OBS analysis tools, empowered by large language models (LLMs), and can flexibly orchestrate these components. Additionally, we construct a comprehensive domain-specific multimodal knowledge base for OBS, which is built through a rigorous multi-year process of data collection, cleaning, and expert annotation. The knowledge base comprises over **1.4M** single-character rubbing images and **80K** interpretation texts. OracleAgent leverages this resource through its multimodal tools to assist experts in retrieval tasks of character, document, interpretation text, and rubbing image. Extensive experiments demonstrate that OracleAgent achieves superior performance across a range of multimodal reasoning and generation tasks, surpassing leading mainstream multimodal large language models (MLLMs) (*e.g., GPT-4o*). Furthermore, our case study illustrates that OracleAgent can effectively assist domain experts, significantly reducing the time cost of OBS research. These results highlight OracleAgent as a significant step toward the practical deployment of OBS-assisted research and automated interpretation systems.

## 1 INTRODUCTION

Oracle Bone Script (OBS) is the earliest known form of the Chinese writing system, dating back more than 3,000 years to the Shang Dynasty (c. 1400–1100 B.C.). Inscribed on turtle plastrons and animal scapulae for divination, ritual, and record-keeping, these inscriptions not only document major historical events and religious practices but also provide invaluable insights into the language, society, and culture of early Chinese civilization, marking a pivotal stage in the evolution of Chinese characters. Despite the discovery of approximately 4,500 OBS characters, only about 1,600 have been successfully deciphered, leaving much of this ancient writing system still undeciphered.

During the decipherment of Oracle Bone Script (OBS), the most fundamental materials to researchers are the approximately 150,000 excavated oracle bone fragments. However, these physical artifacts are dispersed across various locations, making it exceedingly difficult for scholars to access the originals for in-depth study. To overcome this, scholars rely on rubbings, which are paper impressions that capture the surface texture of the bones. As shown in Fig. 1, these rubbings authentically preserve the original information of the oracle bones but often suffer from unclear character shapes due to noise such as scratches. The corpus of rubbings currently amounts to about 200,000 pieces. Additionally, scholars produced facsimiles by manually outlining the oracle bone characters. As illustrated in Fig. 1, these facsimiles feature clear character morphology but inevitably lose some details present on the original fragments. There are approximately 70,000 such traced images that can be directly paired with corresponding rubbings.

Typically, research workflows of OBS involve comparing similar character forms, examining the interpretations of specific character across different rubbings, extracting comprehensive information from duplicate fragments, and synthesizing prior scholarship. However, the absence of Unicode encoding for OBS poses significant challenges for information retrieval. To address this, scholars compiled comprehensive reference works, such as *Oracle Bone Inscriptions Gulin* (Yu, 1996), *Oracle Bone Inscriptions Compendium* (Li, 2012), and *Yinxu Complete Collection of Facsimiles with Transcriptions of Oracle Bone Inscriptions* (Yao, 1998), which serve different purposes: aggregating philological interpretations, cataloging glyph variants, and providing collections of deciphered texts. Despite these efforts, the process has traditionally depended heavily on the expertise and memory of individual scholars. Conducting searches, comparisons, and syntheses across resources at the scale of 200K rubbings, reference works, and academic publications is both time-consuming and error-prone. For instance, even experienced experts may spend considerable time compiling evidence for a single character, while less experienced scholars may require significantly longer. Consequently, the efficiency and accuracy of information retrieval and organization have become critical bottlenecks in OBS research.

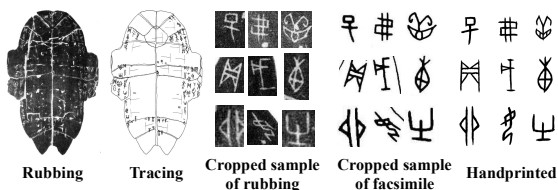

Figure 1: Illustrative examples of oracle bone images from different OBS data modalities.

To address these challenges, we propose OracleAgent, the first agent system designed for the structured management and retrieval of OBS information. OracleAgent is capable of assisting researchers rapidly, accurately, and comprehensively in gathering relevant content and associated information. Our agent is meticulously designed around three core aspects: knowledge base construction, model-based toolchains, and task-oriented planning. This enables the agent to orchestrate tools according to the requirements of specific research tasks, leverage the knowledge base as an enhanced resource for retrieval and generation, and finally aggregate and synthesize all related information for researchers, thereby accelerating the decipherment process of Oracle Bone Script. Its architecture is carefully designed around three core components: **(1) Knowledge Bases:** From a content perspective, the data must comprehensively cover diverse aspects of OBS. We therefore construct five interlinked knowledge bases encompassing rubbings, facsimiles, single characters, interpretation texts, and scholarly literature. From a structural perspective, the heterogeneous combination of images and texts poses challenges for algorithmic processing. To mitigate this, we fragment and restructured resources such as 3,000 research papers and *Gulin* (Yu, 1996) to enable fine-grained retrieval. **(2) Domain Model-Driven Tools:** To enable precise and comprehensive retrieval from the knowledge bases, we develop a suite of algorithms, including single-character detection, glyph retrieval, rubbing retrieval, and facsimile generation. These algorithms not only enrich the knowledge bases through offline processing of raw data but also support online retrieval, ensuring that researchers can directly access the information relevant to their tasks. **(3) Task-Oriented Planning with LLMs:** Powered by LLMs, OracleAgent dynamically plans tasks based on the specific needs of researchers, autonomously selects and invokes the most suitable model-based tools for each subtask, and ultimately aggregates both retrieved and generated information into comprehensive, coherent outputs tailored to the user's requirements.

The main contributions are summarized as: 1) We propose **OracleAgent**, the first AI agent system designed for the structured management and retrieval of OBS information, which seamlessly integrates seven OBS analysis tools, empowered by LLMs, dynamically orchestrating specialized components for complex OBS queries. 2) We propose the first comprehensive, domain-specific multimodal knowledge base for OBS, built through a rigorous multi-year process of data collection, cleaning, and expert annotation. The knowledge base contains over **1.4M** single-character facsimile images and **80K** interpretation texts, supporting retrieval tasks of character, document, interpretation text, and rubbing image via the multiple tools integrated within OracleAgent. 3) OracleAgent provides comprehensive OBS analysis capabilities, including modality classification, character classification and retrieval, character detection, and facsimile generation. It dynamically orchestrates specialized tools based on user needs, retrieves information from knowledge bases, and synthesizes reliable results. 4) Extensive experiments demonstrate that OracleAgent outperforms leading MLLMs on OBS reasoning and generation tasks, while maintaining strong stability. Further case studies show that OracleAgent can effectively execute complex workflows and assist domain experts in retrieving relevant documents, significantly reducing the time required for OBS research.

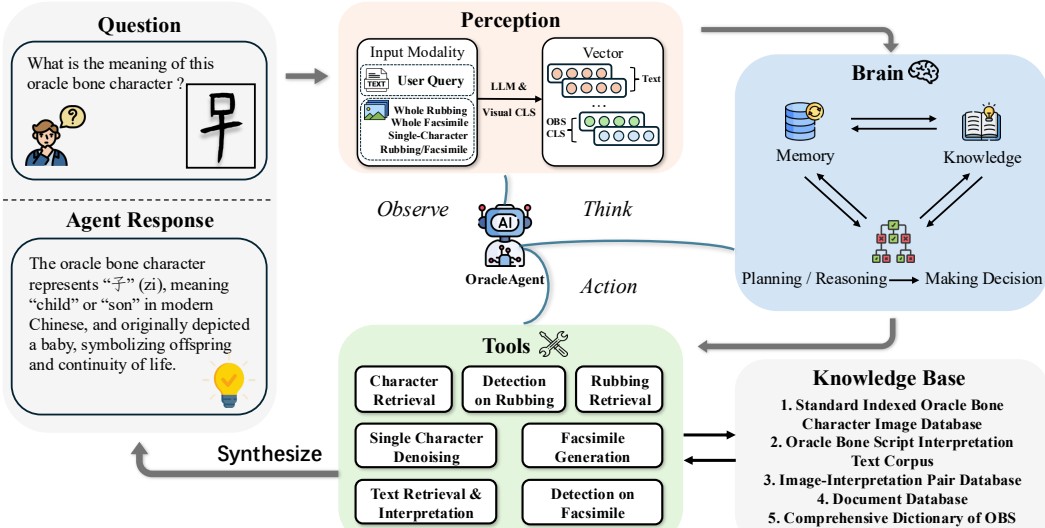

Figure 2: **Architecture overview of the proposed OracleAgent.** OracleAgent consists of four modules: Perception, Brain, Tools, and Knowledge Bases. The Perception module accepts multimodal user inputs and infers user intent. The Brain stores states in Memory and integrates multimodal reasoning with tool-based decision-making. Some tools integrated within OracleAgent are capable of retrieving information from knowledge base.

## 2 RELATED WORKS

**Oracle Character Processing.** Deep learning has played an important role in OBS processing. Recent studies (Jiang et al., 2023; Li et al., 2021; Wang et al., 2022; Gao et al., 2022) have focused on character detection, denoising, and image translation, while others, such as Genov (Qiao et al., 2024) and OracleFusion (Li et al., 2025b), leverage MLLMs to enhance visual understanding. More recently, OBS Decipher (OBSD) (Guan et al., 2024) has applied diffusion models to decipher oracle characters. Although these works are all centered on deep learning, their models are typically tailored to specific tasks and lack a unified framework capable of addressing the full range of OBS challenges. To bridge this gap, OBI-Bench (Chen et al., 2024b) introduces a comprehensive OBS benchmark for MLLMs, leveraging their strong prior knowledge to tackle various OBS tasks. However, due to issues such as hallucination and suboptimal performance of MLLMs on certain tasks, there remains a need for a more comprehensive, effective, and reliable AI system.

**LLM-based Agent Architectures.** Recent advances in LLM-powered agents have enabled autonomous reasoning, planning, and flexible tool utilization (Xi et al., 2025; Zhao et al., 2023; Masterman et al., 2024). Representative frameworks such as ReAct (Yao et al., 2023) combine reasoning and acting, while tool-calling methods (*e.g.,* Toolformer (Schick et al., 2023)) and multi-agent orchestration (*e.g.,* AutoGen (Wu et al., 2024)) further expand agent capabilities. However, their application in domain-specific tasks like OBS research is still underexplored, highlighting the need for customized agent frameworks incorporating domain knowledge and systematic tool coordination.

**Evaluation Frameworks.** A range of benchmarks have been developed to rigorously assess the capabilities of LLM-based agents. For example, AgentBench (Liu et al., 2023) evaluates agents on tasks such as multi-step reasoning, memory retention, tool utilization, task decomposition, and interactive problem solving. Findings indicate that even advanced models like GPT-4o and Claude-3.5-Sonnet face challenges with maintaining long-term context and making autonomous decisions. Building on these insights, MMAU (Yin et al., 2024) extends evaluation to five domains—tool use, graph reasoning, data science, programming, and mathematics—further exposing ongoing difficulties in structured reasoning and iterative problem solving. However, the evaluation of agents in the OBS domain remains unexplored. To address this gap, we compare our approach with general-purpose MLLMs and further extend the benchmark to the domain of facsimile generation.

# 3 ORACLEAGENT

## 3.1 SYSTEM OVERVIEW

We present **OracleAgent**, a unified agent framework designed for various Oracle Bone Script (OBS) tasks. Fig. 2 illustrates the workflow of our OracleAgent, which comprises four core modules: Perception, Brain, Tools, and Knowledge Base. User queries are processed sequentially through these modules, enabling adaptive and context-aware reasoning. The overall workflow is as follows: *(1) Observe*: The Perception module ingests external inputs, including user queries and various types of images, providing a comprehensive understanding of the environment. *(2) Think*: The Brain module dynamically analyzes the current state maintained in Memory and performs structured reasoning by orchestrating an array of specialized tools for decision-making. *(3) Action*: OracleAgent orchestrates multiple tools in serial and parallel workflows to accomplish complex tasks. Furthermore, the integrated multimodal tools within the agent leverage the domain-specific OBS knowledge base for information retrieval, serve as a reliable source for retrieval- augmented generation.

## 3.2 KNOWLEDGE BASES

To better support OBS experts in the organization and study, we construct a comprehensive, richly annotated image database of oracle bone characters together with a corresponding textual-interpretation corpus. This resource is assembled through a multi-year effort involving extensive manual collection, rigorous data cleaning, and meticulous annotation to maximize accuracy and coverage. We further integrate a task-driven image-retrieval framework that leverages the image and text retrieval tools described in Section 3.3. This framework matches input query images of oracle bone characters to canonical forms stored in the database and then retrieves multiple candidate interpretations from the textual corpus, thereby enabling automatic matching of isomorphic characters. The knowledge base comprises five databases. In the first three databases, every image is annotated with provenance information to ensure traceability. In addition, single-character entries, their textual interpretations, and their images are mutually cross-referencable via oracle bone fragment IDs. Moreover, we curat a pixel-level fine-grained dataset of approximately 15K annotations that records facsimile–rubbing correspondences, precise single-character locations, and explicit reading order, providing a high-resolution foundation for downstream retrieval and analysis tasks.

- **Standard Indexed Oracle Bone Character Image Database** (Peichao, 2024): A meticulously curated standard database of oracle bone characters, produced by domain experts, containing 50K images across 4,000 distinct characters and 6,000 subclasses. This database serves as the reference for associating input character images with their standard forms.

- **Oracle Bone Script Interpretation Text Corpus**: A collection of interpretation texts for oracle bone fragments, gathered from online sources, covering 60K fragments and 80K interpretation texts. This corpus includes the meanings of characters, common phrases, and their occurrences in various literature. The main sources include: **(1)** Oracle Bone Instructions (OBI) Collection (Guo & Hu, 1978-1982), **(2)** Supplement to the OBI Collection (Peng et al., 1999) , **(3)** Huayuan East Oracle Bones (Institute of Archaeology, 2003), and **(4)** Xiaotun South Oracle Bones (Liu, 1983).

- **Image-Interpretation Pair Database**: First, we apply the character-detection algorithm described in Section 3.3 to a corpus of 172K rubbings from YinQiWenYuan (AYNU, 2020), yielding 1.4M cropped single-character images. To support efficient retrieval, we then generate facsimile representations for each rubbing and each extracted character by applying the facsimile generation algorithm introduced in Section 3.3. Collectively, these processing steps produce the most comprehensive database of image–interpretation pairs to date.

- **Document Database**: We construct our document database from the YiQinWenYuan, which provides richly interleaved image-text data on oracle bone studies. The database comprises 3,000 documents related to the interpretation of oracle bone characters, alongside three authoritative reference books covering historical perspectives on specific characters. Text regions are extracted from document images using PaddleOCR (Cui et al., 2025), and all character images are manually segmented. Each image is indexed by its corresponding character, enabling precise image-text alignment and facilitating efficient retrieval tasks in downstream modules.

- **Comprehensive Dictionary of OBS**: A database of interpretations for 7,000 oracle bone inscriptions, sourced from *Gulin* (Yu, 1996) and Gulin Supplementary Volume (He, 2017), with the same compilation methodology as the document database.

### 3.3 DOMAIN MODEL TOOLS OF ORACLE BONE SCRIPT

- **Character Detection on Rubbing.** We train a YOLO-based detection model on the rubbing image dataset to automatically localize oracle bone script (OBS) characters within rubbing images.

- **Character Detection on Facsimile.** We train a YOLO-based detection model on the facsimile image dataset to identify and extract oracle bone script (OBS) characters from facsimile images.

- **Text Retrieval and Interpretation.** We utilize the GTE-Qwen2-1.5B (Li et al., 2023) multilingual embedding model instruction-tune on high-quality query–document pairs to retrieve and interpret relevant textual information in response to user queries.

- **Character Retrieval and Classification.** To enable efficient character retrieval and classification based on visual similarity, we train a feature extraction model (Ren et al., 2022) specifically designed for characteristics of OBS facsimile images.

- **Single-Character Facsimile Image Denoising.** To transform noisy rubbing images of individual OBS characters into clean facsimile representations, we employ and train a CycleGAN (Zhu et al., 2017) model for this image-to-image translation task.

- **Whole Facsimile Image Generation.** To generate complete facsimile images from rubbing images. We train a ControlNet (Zhang et al., 2023) based on SD1.5 (Rombach et al., 2022) on the OBIMD dataset (Li et al., 2024) to achieve facsimile image generation.

- **Rubbing Image Retrieval.** We support efficient retrieval of rubbing images based on visual similarity by training a specialized matching model (Li et al., 2025a). Furthermore, by leveraging the Image-Interpretation Pair Database, the retrieved rubbing images can be indexed to their corresponding interpretation texts, enabling effective rubbing-to-interpretation matching.

### 3.4 OBS MULTIMODAL PERCEPTION AND BRAIN

The Perception module constitutes the foundational component of our Oracle Bone Intelligent Agent, enabling comprehensive understanding of both user queries and multiple visual modalities of OBS. As illustrated in Fig. 1, The system processes a set of oracle bone images $I = \{I_1, ..., I_i\}_{i=M}$ spanning modalities such as rubbings, facsimiles, single-character crops, and handprinted characters. Each image $I_i$ is encoded via a visual encoder $\mathcal{V}$ to obtain modality-specific features:

$$\mathbf{v}_i = \mathcal{V}(I_i), i = 1, 2, ..., M. \tag{1}$$

User queries $Q$ are interpreted in the context of the visual inputs. Unlike conventional approaches that combine multi-modal features via concatenation or attention mechanisms, our method employs large language models (LLMs) by injecting discrete visual tokens into the textual prompt. Specifically, visual features $\{\mathbf{v}_1, \mathbf{v}_2, ..., \mathbf{v}_i\}_{i=M}$ are projected into tokens and embedded alongside the user query to form a unified prompt, which can be formulated as:

$$Prompt = [Q; \mathbf{v}_1; \mathbf{v}_2; ...; \mathbf{v}_i]_{i=M} \tag{2}$$

This unified prompt enables the Agent to jointly process textual and visual information, facilitating deep semantic alignment and cross-modal reasoning through the LLM's contextual capabilities.

The Brain module serves as the central reasoning and decision-making component of the Agent, functioning as the "cognitive core" that orchestrates the overall workflow. Leveraging the powerful prior knowledge embedded within large language models (LLMs), the Brain module is responsible for analyzing the current state stored in Memory, planning tool usage, performing multi-step reasoning, and ultimately making decisions to fulfill user intents. At each interaction step, the Agent maintains a dynamic state $s_t$ in Memory, which encapsulates the historical context, user queries, intermediate results, and relevant environmental information up to time $t$. The Brain module first retrieves and interprets this state: $s_t = Memory(t)$. The state $s_t$ is then encoded into a structured prompt, which is fed into the LLM-based Brain for further analysis. Given the current state $s_t$, the Brain module utilizes its extensive prior knowledge to plan the sequence of tool invocations required to solve the task. Let $\mathcal{A} = \{a_1, a_2, ...a_j\}_{j=K}$ denote the set of available tools (*e.g., OCR, image*

Table 1: Results on the OBS character retrieval task on OBC306 and OBI-IJDH datasets. *"Yes-or-No"* and *"How"* represent the absolute and probability output, respectively. We report the averaged Recall@1, 3, 5, and mAP@5 for *"How"* question.

| Models | OBC306 | | | | OBI-IJDH | | | |
|---|---|---|---|---|---|---|---|---|
| | *Yes-or-No*↑ | *How*↑ | | | *Yes-or-No*↑ | *How*↑ | | |
| | mAP@\|Yes\| | Recall@1 | Recall@3 | mAP@5 | mAP@\|Yes\| | Recall@1 | Recall@3 | mAP@5 |
| GPT-4v | 0.4228 | 0.205 | 0.650 | 0.624 | 0.5680 | 0.225 | 0.676 | 0.740 |
| GPT-4o | 0.4550 | **0.235** | 0.686 | 0.688 | 0.6122 | 0.250 | 0.706 | 0.800 |
| Qwen-VL-MAX | 0.4223 | 0.190 | 0.621 | 0.644 | 0.5716 | 0.225 | 0.638 | 0.780 |
| InternVL2-Llama3-76B | 0.3557 | 0.150 | 0.460 | 0.522 | 0.4268 | 0.250 | 0.675 | 0.720 |
| InternVL2-8B | 0.2844 | 0.095 | 0.374 | 0.420 | 0.3623 | 0.225 | 0.650 | 0.68 |
| Qwen-VL-7B | 0.2883 | 0.080 | 0.345 | 0.422 | 0.3528 | 0.225 | 0.588 | 0.660 |
| LLaVA-NeXT-8B | 0.2793 | 0.075 | 0.358 | 0.348 | 0.3605 | 0.225 | 0.606 | 0.600 |
| Qwen2.5-VL-7B | 0.2995 | 0.110 | 0.388 | 0.460 | 0.3704 | 0.225 | 0.620 | 0.700 |
| **OracleAgent (Ours)** | **0.4953** | 0.210 | **0.690** | **0.712** | **0.7600** | 0.250 | **0.735** | **0.940** |

*retrieval, translation, etc*). At time $t$, the Brain constructs a plan $\pi_t$, which is an ordered sequence of tool actions: $\pi_t = [a_1, a_2, ..., a_n], a_i \in \mathcal{A}$, which maximizes the expected utility:

$$\pi_t = \arg\max_\pi \mathbb{E}[R(\pi \mid s_t, \text{Goal})], \tag{3}$$

where $R(\cdot)$ denotes the utility of executing plan $\pi$ given the state $s_t$ and the user goal.

## 4 ORACLEAGENT-BENCH

### 4.1 DATASET

For OBS detection, character classification, and character retrieval tasks, we use OBI-Bench (Chen et al., 2024b) to evaluate the understanding and reasoning ability of OracleAgent. Additionally, we introduce 3K images from 4 different OBS modalities illustrated in Fig. 1 to evaluate the OBS modality classification and facsimile generation ability.

Table 2: Results on the OBS detection task. *"How"* and *"Where"* represent the number and bounding box output, respectively. We report MRE and mIoU.

| Models | Type | *How*↓ | *Where*↑ |
|---|---|---|---|
| GPT-4v | Closed | 0.4383 | 0.0165 |
| GPT-4o | Closed | **0.3458** | 0.0182 |
| Qwen-VL-MAX | Closed | 0.4843 | 0.0131 |
| InternVL2-Llama3-76B | Open | 0.5344 | 0.0623 |
| InternVL2-8B | Open | 1.1146 | 0.0152 |
| Qwen-VL-7B | Open | 3.5694 | 0.0069 |
| LLaVA-NeXT-8B | Open | 0.4268 | 0.0189 |
| Qwen2.5-VL-7B | Open | 1.0000 | 0.1112 |
| **OracleAgent (Ours)** | Open | 0.3894 | **0.6198** |

### 4.2 QUESTION SETTINGS

To evaluate various perception capabilities, we follow the coarse-to-fine question settings in OBI-Bench. Specifically, we categorize the questions into four distinct types: **(1)** ***Yes-or-No***: Binary questions designed to minimize ambiguity and directly reflect the underlying task objectives. **(2)** ***Which***: Single-choice questions that require the model to identify the correct answer from a finite set of candidates, thereby evaluating its discriminative capability among closely related options. **(3)** ***How***: Quantitative questions, such as determining the number of oracle bone characters present in an image or estimating the probability that two characters belong to the same class, which facilitate a more fine-grained assessment of model performance. **(4)** ***Where***: Localization questions that prompt the model to output bounding boxes for detected characters, thereby assessing its spatial reasoning and detailed perceptual abilities. For facsimile generation task, we prompt the model with instruction: "Please transform this picture into a facsimile.".

## 5 EXPERIMENTS

### 5.1 IMPLEMENTATIONS

OracleAgent employs DeepSeek-V3.1 (Liu et al., 2024a) as its backbone LLM and integrates Yolov11 (Khanam & Hussain, 2024) for character detection, GTE-Qwen2-1.5B (Li et al., 2023)

Table 3: Results on the OBS character classification task on HWOBC, Oracle50K, and OBI125 datasets. Note that *"Yes-or-No"* and *"How"* represent absolute and probability output, respectively.

| Models | HWOBC | | | Oracle-50k | | | OBI125 | | |
|---|---|---|---|---|---|---|---|---|---|
| | *Yes-or-No*↑ | *How*↑ | | *Yes-or-No*↑ | *How*↑ | | *Yes-or-No*↑ | *How*↑ | |
| | Acc | Acc@1 | Acc@5 | Acc | Acc@1 | Acc@5 | Acc | Acc@1 | Acc@5 |
| GPT-4v | 69.50 | 86.75 | **100.0** | 66.00 | 88.25 | **100.0** | 57.75 | 70.75 | 91.75 |
| GPT-4o | 72.75 | 89.75 | **100.0** | 74.50 | 90.25 | **100.0** | 62.50 | 75.50 | 93.75 |
| Qwen-VL-MAX | 64.25 | 85.00 | **100.0** | 65.75 | 88.75 | 98.75 | 55.00 | 69.25 | 89.75 |
| InternVL2-Llama3-76B | 44.75 | 53.75 | 69.75 | 47.50 | 55.00 | 69.00 | 43.25 | 50.75 | 66.75 |
| Qwen-VL-7B | 44.25 | 48.00 | 61.25 | 42.00 | 51.00 | 63.50 | 38.75 | 44.75 | 61.25 |
| InternVL2-8B | 42.25 | 47.75 | 59.75 | 41.75 | 49.00 | 59.00 | 38.75 | 47.75 | 56.50 |
| LLaVA-NeXT-8B | 44.00 | 46.75 | 53.75 | 42.00 | 46.25 | 56.75 | 38.75 | 42.75 | 54.75 |
| Qwen2.5-VL-7B | 45.50 | 49.25 | 65.25 | 43.25 | 51.75 | 65.25 | 42.50 | 47.25 | 62.75 |
| **OracleAgent (Ours)** | **89.75** | **95.75** | **100.0** | **90.25** | **92.75** | **100** | **80.50** | **84.75** | **95.00** |

for text retrieval and interpretation, EGFF model (Ren et al., 2022) for Glyph Retrieval and Classification, CycleGAN (Zhu et al., 2017) and ControlNet (Zhang et al., 2023) for facsimile generation. OracleAgent executes tool operations via structured JSON API calls, explicitly specifying all required parameters (*e.g., image file locations, textual instructions*) for each target tool. For baseline comparisons, we use the official implementations of all models and strictly follow their recommended configuration protocols during evaluation. For model responses, we employ regular expressions to extract answers such as numerical values or boolean results. In cases of errors or timeouts, the extraction procedure is retried up to three times. If the response remains invalid or does not yield a single definitive answer after these attempts, it is marked as incorrect.

## 5.2 EXPERIMENTAL SETUP

We evaluate OracleAgent against both mainstream open-source and proprietary MLLMs, including the Qwen (Bai et al., 2023), GPT-4 (Hurst et al., 2024), InternVL (Chen et al., 2024a), and LLaVA (Liu et al., 2024b) series. Since most MLLMs lack the advanced instruction-based image editing capabilities like GPT-4o, we additionally compare OracleAgent with Bagel (Deng et al., 2025) and Step1x-Edit (Liu et al., 2025) on the facsimile generation task. We conduct comprehensive evaluations on OracleAgent-Bench, which incorporates experimental settings from OBI-Bench for oracle bone script (OBS) detection, character classification, and retrieval tasks. For modality classification and generation tasks, we design tailored evaluation procedures and metrics. The experimental configurations for the five key domain problems are detailed as follows:

**Character Retrieval and Classification.** To evaluate model performance on oracle bone character retrieval and classification tasks, we design *"Yes-or-No"* and *"How"* questions. Specifically, these questions are constructed based on intra-class and inter-class pairs of oracle bone character images, requiring the model to output either a binary decision or a probabilistic score indicating class similarity. The retrieval task includes 600 images sampled from OBC306 (Huang et al., 2019) and OBI-IJDH (Fujikawa & Meng, 2020). We employ averaged Recall@k and mean Average Precision (mAP) to quantify the multi-round OBI retrieval performance of MLMMs and OracleAgent. The character classification task comprises 500 images across 100 categories from each of HWOBC (Li et al., 2020), Oracle-50k (Han et al., 2020), and OBI125 (Yue et al., 2022), with accuracy (Acc) used as the evaluation metric.

**Detection.** To assess model performance on OBS detection, we employ two question types to evaluate coarse- and fine-grained perceptual abilities with 2K OBS rubbing images sampled from YinQi-WenYuan (AYNU, 2020). *"How"* questions require the model to predict the number of characters on a given rubbing, while *"Where"* questions task the model with precisely localizing each character by outputting its bounding box. We utilize MRE described in Eq. 4 and mIoU

**Modality Classification.** We utilize 2K images from OBIMD (Li et al., 2024), covering four distinct OBS image modalities. For each image, we present a *"Which"* question, requiring the model to select the most appropriate modality from four given options.

**Generation.** The facsimile generation task converts oracle bone rubbing images into corresponding facsimiles via a simple prompt (*e.g., "Please convert this to a facsimile."*). Most existing MLLMs lack the instruction-driven image editing capabilities of models like GPT-4o. We evaluate OracleAgent against unified autoregressive models, including Bagel (Deng et al., 2025), Flux1.-

Kontext (Batifol et al., 2025), and Step1x-Edit (Liu et al., 2025). We sample 500 rubbing-facsimile pairs from designated test set of the OBIMD dataset and assess performance using standard metrics: FID (Heusel et al., 2017), KID (Bińkowski et al., 2018), SSIM (Wang et al., 2004), and LPIPS (Zhang et al., 2018). For further details, please refer to the OBI-Bench paper (Chen et al., 2024b) and Appendix E of our work.

**Case Study of Retrieval Evaluation.** For the retrieval task of a specific oracle character, we use expert-annotated results as the ground truth and compare them with the retrieval results obtained by OracleAgent. Specifically, we evaluate the retrieval performance using Precision, Recall, F1-score, and Coverage, with their respective definitions and calculation formulas provided in Eq. 5-Eq. 8.

## 5.3 QUANTITATIVE ANALYSIS

**Retrieval.** As shown in Tab. 1, all models exhibit comparable performance on the character retrieval task under the Recall@1 metric, with OracleAgent slightly trailing GPT-4o on the OBC306 dataset. However, when expanding the candidate pool to Recall@3 and Recall@5, OracleAgent consistently outperforms all baselines on both the OBC306 and OBI-IJDH datasets across *"Yes-or-No"* and *"How"* question types. This indicates that OracleAgent is more

Table 4: Results on the OBS modality classification task.

| Models | Type | Which↑ | | |
|---|---|---|---|---|
| | | Acc@1 | Precision | Recall |
| Qwen-VL-MAX | Closed | 83.35 | 0.8731 | 0.8345 |
| Qwen-VL-PLUS | Closed | 70.40 | 0.8561 | 0.7050 |
| InternVL2-8B | Open | 65.95 | 0.7748 | 0.6595 |
| Qwen-VL-7B | Open | 35.45 | 0.3734 | 0.2665 |
| LLaVA-NeXT-8B | Open | 39.55 | 0.4465 | 0.2955 |
| Qwen2.5-VL-7B | Open | 43.75 | 0.5434 | 0.4375 |
| **OracleAgent (Ours)** | Open | **99.90** | **0.9990** | **0.9975** |

effective at retrieving relevant characters when a broader set of candidates is considered, which is critical for practical applications. These findings also underscore the limitations of Recall@1 in distinguishing model capabilities. Moreover, OracleAgent achieves the highest mAP@5 scores on both datasets, further demonstrating its superior retrieval performance and robustness in complex character retrieval scenarios compared to state-of-the-art models.

**Character Classification.** Tab. 3 presents OBS character classification results on HWOB, Oracle-50k, and OBI125. OracleAgent consistently achieves the highest accuracy on both *"Yes-or-No"* and *"How"* questions across all datasets. While proprietary models such as Qwen and GPT-4 outperform open-source MLLMs, OracleAgent maintains a clear lead, particularly in Acc@5. These results demonstrate its superior classification and generalization capabilities.

**Detection.** As shown in Tab. 2, OracleAgent achieves the highest performance on the *"Where"* (mIoU) metric with a score of 0.6198, significantly outperforming all baselines and demonstrating strong localization capability. On the *"How"* (MRE) metric, OracleAgent attains 0.3894, slightly higher than GPT-4o but better than most open models, indicating robust numerical prediction. Overall, OracleAgent substantially improves target localization accuracy while maintaining low quantity prediction error, highlighting its superior fine-grained perception.

**Modality Classification.** As shown in Tab. 4, OracleAgent achieves an accuracy of 99.9% on the modality classification task. Since modality classification is a fundamental step upon which subsequent workflows depend, such high accuracy is crucial. Moreover, other models lack access to multimodal OBS training data, which reasonably accounts for their lower zero-shot performance.

**Evaluation on Generation** As shown in Tab. 5, OracleAgent outperforms all baselines on the OBS facsimile generation task across all evaluation metrics. OracleAgent produces facsimiles that are both visually and perceptually closer to the ground truth, with FID and KID reduced by more than half com-

Table 5: Results on the OBS facsimile generation task.

| Models | FID↓ | KID↓ | SSIM↑ | LPIPS↓ |
|---|---|---|---|---|
| GPT-4o | 103.65 | 68.19 | 0.8127 | 0.2787 |
| Step1x-Edit | 303.93 | 292.56 | 0.5876 | 0.4853 |
| Flux.1-Kontext | 225.88 | 196.73 | 0.7232 | 0.3491 |
| Bagel-7B | 144.02 | 95.97 | 0.7087 | 0.3417 |
| **OracleAgent (Ours)** | **45.06** | **24.00** | **0.8381** | **0.2447** |

pared to GPT-4o, and notable gains in SSIM and LPIPS. These results highlight OracleAgent's advanced fine-grained facsimile image generation capability.

### 5.4 CASE STUDY

#### 5.4.1 INTERACTION FLOW

As shown in Fig. 3, the user's question concerns the analysis of a given oracle bone rubbing. OracleAgent first recognizes the modality of the OBS rubbing image. Next, it automatically invokes the object detection tool to accurately identify all oracle bone characters present on the rubbing. Using the rubbing retrieval tool, it then determines the fragment number, thereby obtaining the correct reading order of the characters. Finally, by calling the interpretation retrieval tool, OracleAgent retrieves the corresponding modern Chinese characters and aligns each oracle bone character with its modern counterpart. This process simulates the workflow of OBS expert and can greatly simplify their work. The subsequent interaction is displayed in Fig. 4.

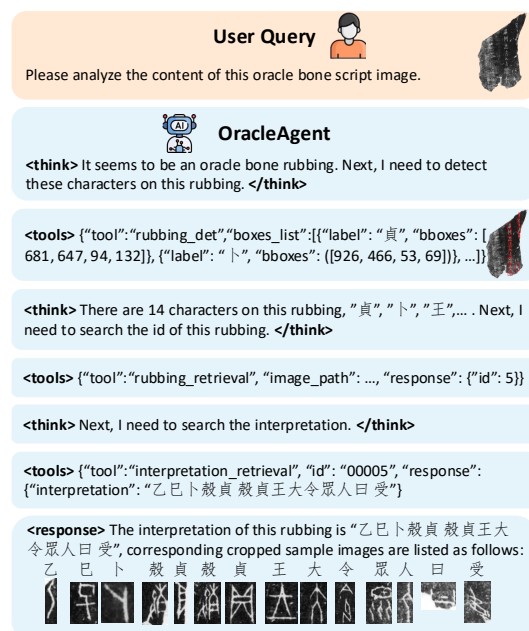

Figure 3: **OracleAgent Interaction Flow:** Automated analysis of an oracle bone rubbing

Table 6: Quantitative retrieval result for a specific oracle bone character.

| Precision ↑ | Recall ↑ | F1-score ↑ | Coverage ↑ |
|---|---|---|---|
| 0.8959 | 0.9269 | 0.9112 | 0.9615 |

#### 5.4.2 RETRIEVAL EVALUATION

To comprehensively assess the performance of OracleAgent in real-world OBS information retrieval tasks, we conducted a comparative experiment against domain experts' manual search results. Specifically, we selected a target oracle character and tasked both OracleAgent and human experts with identifying all oracle bone fragments in which the character appears. For the expert baseline, a team of domain specialists engaged in manual retrieval by consulting relevant reference books and materials over a one-week period. The aggregated expert findings serve as the ground truth (noting that these results might not be fully exhaustive, but currently represent the best available benchmark). OracleAgent subsequently performed the same retrieval task automatically, and its results were directly compared with those from the experts. As shown in Tab. 6, OracleAgent demonstrated strong performance on this task, achieving a high recall of 92.69% and category coverage of 96.15%. These metrics indicate a high level of agreement between OracleAgent and expert results. Additionally, OracleAgent retrieved 7.31% more potential instances than the experts, with the majority of these additional findings validated as reasonable upon further expert review. This highlights OracleAgent's expert-level retrieval capabilities and its potential to outperform manual expert searches, thus providing a robust foundation for accelerating information retrieval and research in oracle bone studies.

## 6 CONCLUSION

In this work, we introduce OracleAgent, a pioneering AI agent system for the structured management and retrieval of OBS information. OracleAgent addresses two long-standing challenges in OBS research: the complexity of interpretation workflows and inefficiencies in information organization and retrieval. By seamlessly integrating multiple OBS model-driven tools via large language models (LLMs) and orchestrating them flexibly, OracleAgent enables end-to-end support for expert tasks. Our comprehensive OBS multimodal knowledge base, comprising over 1.4 million character rubbing images and 80K interpretation texts, substantially enhances the system's capabilities. Experimental results and case studies demonstrate that OracleAgent achieves state-of-the-art performance and significantly reduces the time required for OBS research. Our findings underscore OracleAgent as an important advance toward intelligent, automated support in OBS research. Looking ahead, we plan to further expand the coverage of the knowledge base and explore adaptive agent strategies for broader semiotic and historical domains. We anticipate that OracleAgent will serve as a foundation for future progress in computational humanities and the digitization of ancient scripts.

# 7 REPRODUCIBILITY STATEMENT

We have already elaborated on all the models or algorithms proposed, experimental configurations, and benchmarks used in the experiments in the main body or appendix of this paper. Furthermore, we declare that the entire code used in this work will be released after acceptance.

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

## A    APPENDIX

## B    THE USE OF LARGE LANGUAGE MODELS

We use large language models solely for polishing our writing, and we have conducted a careful check, taking full responsibility for all content in this work.

## C    ETHICS STATEMENT

In alignment with the ICLR Code of Ethics, our work centers on publicly available, open-licensed images of oracle bone script. No sensitive, private, or personally identifiable information has been incorporated at any stage of dataset development or annotation. We exercised particular care to respect cultural heritage and maintain fairness throughout the entire process. To promote transparency and replicability within oracle bone script research, we commit to sharing our dataset and algorithmic resources for academic use only. By doing so, we hope to advance responsible research practices and contribute positively to the scholarly study of ancient scripts.

## D    ADDITIONAL CASE STUDY

In Section 5.4 of the main paper, we present only a portion of the results from our case study. The subsequent user query. In the subsequent interaction, the user follows up on the previous response by inquiring about a specific oracle bone character identified in the initial analysis in Fig. 4. This scenario reflects a typical expert workflow, where further investigation is required for a particular character of interest. OracleAgent recognizes that the query pertains to a single character cropped from an oracle bone rubbing. Given that our Standard Index Database is organized by facsimile modality, OracleAgent first performs single-character denoising to generate the facsimile form of the queried character. It then utilizes the character retrieval tool to locate the standard glyph within the index database and further retrieves visually similar glyphs from the Image-Interpretation

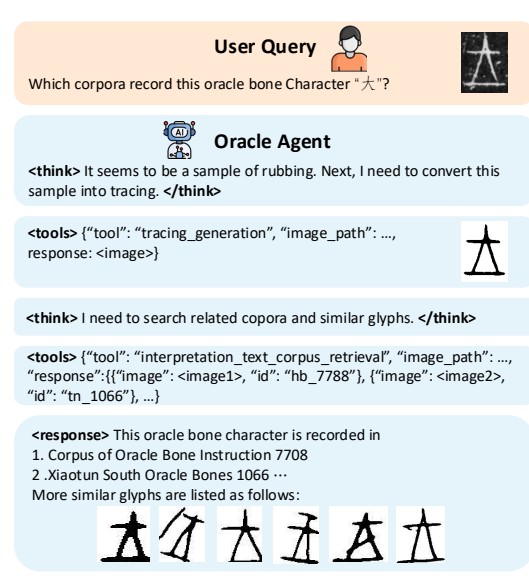

Figure 4: **OracleAgent Interaction Flow:** Follow-up query about which catalogues record this oracle bone character from the last response.

Pair Database, linking each to the corresponding interpretation and corpus.

# E    DETAILED EXPERIMENTAL CONFIGURATION

In this section, we provide a comprehensive description of the experimental setup, including a diverse set of OBS tasks. Specifically, we exhibit the question templates of different oracle bone script tasks.

## E.1    CHARACTER RETRIEVAL AND CLASSIFICATION

***"How"* question template:**
*#System: You are a senior oracle bone researcher who excels in classifying oracle bone characters.*
*#User: Given the following two oracle bone characters, estimate the probability that they belong to the same class. Please return only a single integer between 0 and 100. <image1> <image2>*
***"Yes-or-No"* question template:**
*#System: You are a senior oracle bone researcher who excels in classifying oracle bone characters.*
*#User: Whether these two oracle bone characters belong to the same class? Please return "Yes" or "No". <image1> <image2>*

## E.2    MODALITY CLASSIFICATION

***"Which"* question template:**
*#System: You are a senior oracle bone researcher who excels in classifying the modality of oracle bone images.*
*#User: Which modality is this oracle bone image belong to? <image1>*
*A. Whole Rubbing Image.*
*B. Whole Facsimile Image.*
*C. Single Character Rubbing Image.*
*D. Single Character Facsimile Image.*

## E.3    DETECTION

***"How"* question template:**
*#System: You are a senior oracle bone researcher who excels in detecting characters on oracle bone script images.*
*#User: How many oracle bone characters are in this image? Please return the number of oracle bone characters in this image. <image1>*
***"Where"* question template:**
*#System: You are a senior oracle bone researcher who excels in detecting characters on oracle bone script images.*
*#User: How many oracle bone characters are in this image? For each detected oracle bone character, please return a bounding box in [xmin, ymin, xmax, ymax] format. <image1>*
For *"How"* questions, we employ the relative counting error (MRE) metric to evaluate the performance of different LMMs:

$$\text{MRE} = \frac{1}{N} \sum_{i=1}^{N} \frac{|K_i^{gt} - K_i^{pre}|}{C_i^{gt}}, \tag{4}$$

where $N$ is the number of evaluated OBS images. $K_i^{gt}$ and $K_i^{pre}$ represent the ground-truth and predicted numbers of oracle bone characters in the $i$-th OBS image, respectively.

## E.4    GENERATION

For the facsimile image generation task, the input to our framework is a high-resolution rubbing image of an oracle bone inscription, which captures the raw visual appearance of the engraved characters along with the noise and texture artifacts introduced by the rubbing process. The objective

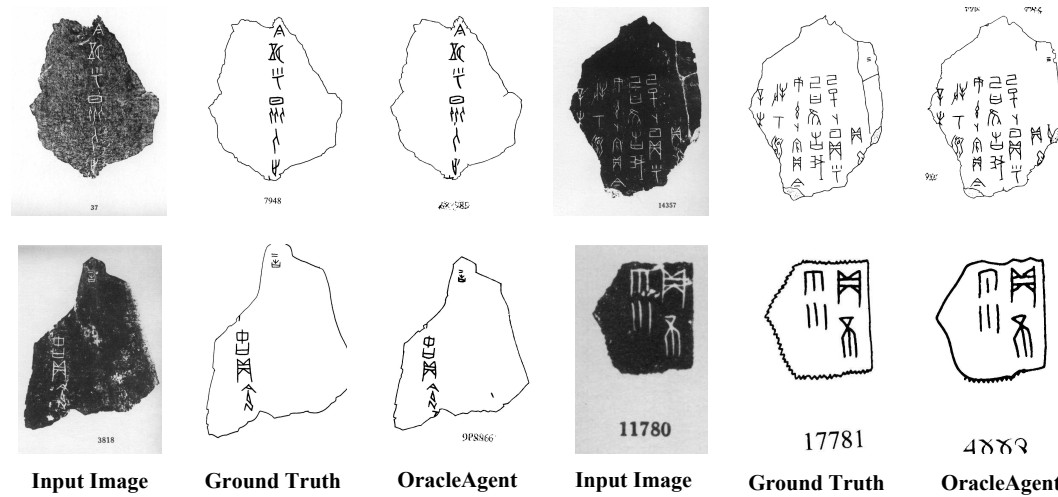

| Input Image | Ground Truth | OracleAgent | Input Image | Ground Truth | OracleAgent |

Figure 5: Examples of facsimile image generation. Note that input image is a rubbing image and OracleAgent generate its facsimile form.

is to transform this noisy archaeological input into a clean, interpretable facsimile representation suitable for scholarly analysis and publication.

As illustrated in Fig. 5, we present four representative examples of the complete generation pipeline. In each case, the left column shows the original rubbing image obtained from an actual oracle bone artifact. The middle column contains the ground-truth hand-drawn facsimile meticulously created by an epigraphic expert, which serves as the gold standard for accurate stroke reconstruction and character interpretation. The right column displays the facsimile automatically generated by OracleAgent, which aims to replicate the expert's style and accuracy while removing background noise, cracks, and irrelevant visual artifacts.

This side-by-side comparison highlights OracleAgent's ability to produce high-fidelity facsimiles that are visually and semantically consistent with expert renderings. The generated results not only preserve the structural integrity and stylistic characteristics of the oracle bone script but also significantly reduce the manual workload traditionally required for such tasks, thereby enabling scalable and efficient digital epigraphy.

## F    RETRIEVAL METRICS OF CASE STUDY

Given a set of ground truth results and predicted results for a specific retrieval task, the evaluation metrics are defined as follows:

**Precision** measures the proportion of correctly retrieved items among all retrieved items:

$$\text{Precision} = \frac{TP}{TP + FP} \tag{5}$$

where $TP$ (True Positive) is the number of relevant items correctly retrieved, and $FP$ (False Positive) is the number of irrelevant items incorrectly retrieved.

**Recall** measures the proportion of correctly retrieved items among all relevant items:

$$\text{Recall} = \frac{TP}{TP + FN} \tag{6}$$

where $FN$ (False Negative) is the number of relevant items that were not retrieved.

**F1-score** is the harmonic mean of Precision and Recall:

$$\text{F1-score} = \frac{2 \times \text{Precision} \times \text{Recall}}{\text{Precision} + \text{Recall}} \tag{7}$$

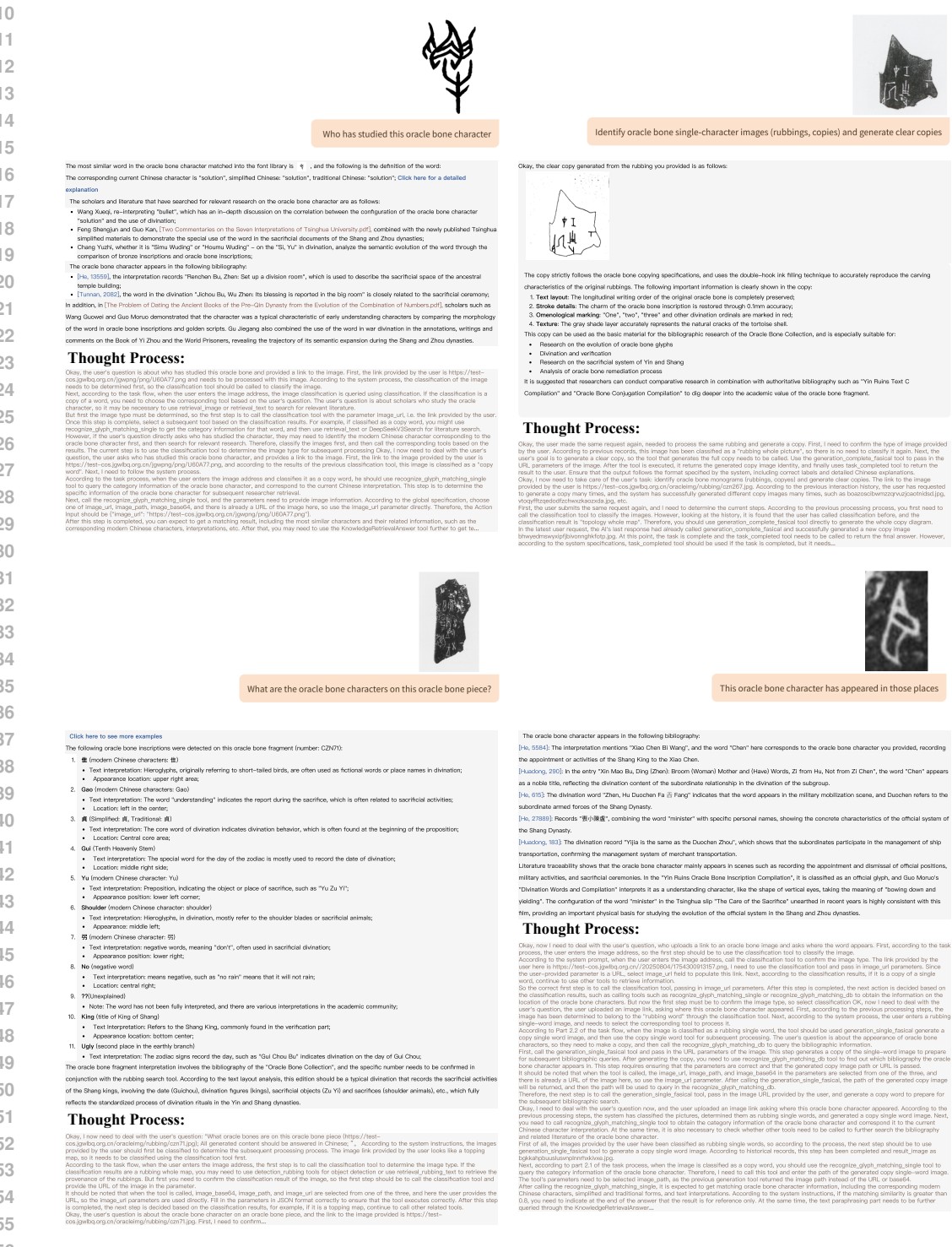

Figure 6: Additional representative user case studies illustrating OracleAgent's ability to process diverse oracle bone script inputs and user queries.

**Coverage** measures the proportion of ground truth categories that are covered by the predicted results, regardless of the count of occurrences. Specifically, for the retrieval of oracle bone characters, Coverage reflects whether a predicted result hits a true category at least once, without considering how many times it is hit (e.g., if an oracle bone character appears multiple times on a single

fragment, only its presence is considered):

$$\text{Coverage} = \frac{|\text{Pred} \cap \text{Real}|}{|\text{Real}|} \tag{8}$$

where Pred is the set of predicted items and Real is the set of ground truth items.

## G  MORE EXAMPLES OF CASES STUDY.

As shown in Fig. 6, we present representative user case studies illustrating OracleAgent's ability to process diverse oracle bone script (OBS) inputs and queries. Inputs include rubbing images, cropped single-character segments, and cropped single-character facsimile, accompanied by user requests such as character identification, semantic interpretation, document retrieval, and clean facsimile generation. For each case, we show the user query, the system's response, and the reasoning trace that reveals OracleAgent's step-by-step decision process. These examples highlight the system's versatility across various oracle bone modalities and its capability to deliver accurate, interpretable results for epigraphic research.

In summary, these case studies demonstrate that OracleAgent can effectively handle a wide range of input modalities and user queries in oracle bone script research. The system's ability to provide accurate, interpretable, and context-aware responses highlights its potential to facilitate and accelerate epigraphic analysis. We believe OracleAgent represents a significant step toward the development of comprehensive AI-assisted tools for historical document interpretation.

