# OpenReview forum: "OracleAgent: A Multimodal Reasoning Agent for Oracle Bone Script Research"
_ICLR.cc/2026/Conference — ICLR 2026 Conference Withdrawn Submission_

### Official Review · Reviewer_qcZm · 2025-10-29

**Soundness:** 3
**Presentation:** 3
**Contribution:** 3
**Rating:** 6
**Confidence:** 2

**Summary:**

This paper introduces OracleAgent, a multimodal AI agent system designed to address key challenges in Oracle Bone Script (OBS) research, such as the complexity of interpretation workflows and inefficiencies in information retrieval.
OracleAgent integrates seven domain-specific tools powered by LLMs and a meticulously constructed multimodal knowledge base to enable structured management and retrieval.
The system operates through a modular pipeline: Perception, Brain, Tools, and Knowledge Bases. Experiments on tasks like character classification, detection, and facsimile generation demonstrate that OracleAgent outperforms state-of-the-art MLLMs.

**Strengths:**

1. OracleAgent combines LLMs with domain-specific tools (e.g., YOLO-based detection and CycleGAN for denoising) to handle complex, multi-step workflows. The agent’s dynamic planning capability allows flexible tool orchestration based on user queries, simulating expert reasoning.
2. The construction of a large-scale, multimodal OBS knowledge base is a significant contribution. It supports fine-grained retrieval across rubbings, facsimiles, and texts, addressing critical bottlenecks in resource accessibility.
3. The paper includes extensive experiments on diverse tasks (e.g., retrieval, classification, generation) using established benchmarks (e.g., OBI-Bench) and custom metrics.
4. Case studies (e.g., automated rubbing analysis and character retrieval) validate real-world utility.

**Weaknesses:**

1. The LLM-driven task planning process is opaque. For example, the paper does not explain how OracleAgent prioritizes tools (e.g., why it chooses denoising before retrieval for a cropped character) or how experts can verify/debug its decisions.
2. While experiments cover standard tasks, there is no analysis of OracleAgent’s performance on severely damaged rubbings or undeciphered characters.
3. While case studies simulate expert workflows, there is no feedback from actual OBS scholars (e.g., usability surveys, task completion time comparisons with human experts). This limits insights into how well OracleAgent aligns with real scholarly practices.

**Questions:**

See Weaknesses.

---

### Official Review · Reviewer_jJAQ · 2025-10-30

**Soundness:** 3
**Presentation:** 4
**Contribution:** 3
**Rating:** 6
**Confidence:** 3

**Summary:**

This paper introduces OracleAgent, the first multimodal reasoning agent tailored for Oracle Bone Script (OBS) research — one of the earliest known writing systems in ancient China. The paper identifies two major challenges in OBS studies: (1) the highly complex and fragmented workflow of character interpretation, and (2) inefficient information organization across dispersed multimodal resources.

OracleAgent tackles these issues through a unified agent framework that integrates large language model (LLM)–based reasoning and planning with a suite of domain-specific visual and textual tools. The system operates through four key modules: Perception, Brain, Tools, and Knowledge Bases, enabling dynamic orchestration of model-driven components.

The authors also build a comprehensive OBS knowledge base containing 1.4 M single-character images and 80 K interpretation texts, linked across rubbings, facsimiles, characters, and textual corpora. OracleAgent is evaluated via OracleAgent-Bench, extending OBI-Bench with new tasks (character retrieval, classification, detection, modality classification, facsimile generation). It outperforms GPT-4o, Qwen-VL, and InternVL baselines in both retrieval and generation metrics.

**Strengths:**

Novel System Integration: First multimodal agent framework unifying LLM reasoning with domain-specific tools for ancient script decipherment.

Rich Multimodal Knowledge Base: Large, expertly annotated dataset linking images and texts across multiple sources.

Comprehensive Evaluation: Benchmarks across five major tasks with quantitative superiority over GPT-4o and other MLLMs.

High Practical Value: Demonstrated ability to replicate and accelerate expert workflows; clear cultural and scientific significance.

Strong Presentation: Detailed figures and step-by-step case studies clarify the pipeline and results.

**Weaknesses:**

Limited Algorithmic Novelty: Most modules rely on pre-existing models; the innovation lies mainly in orchestration rather than method.

Reproducibility Details: Some implementation details (training hyperparameters, LLM fine-tuning procedures, tool-calling logic) are missing.

Domain Generalization: It remains unclear how easily OracleAgent can be transferred beyond OBS to other semiotic or multimodal domains.

Ablation Studies: The paper would benefit from an analysis isolating the contribution of each component (e.g., LLM planning vs. domain tools).

Evaluation Bias: Expert comparison is convincing but somewhat small-scale; broader human evaluation would strengthen the claim of “expert-level retrieval.”

**Questions:**

1- How does the LLM planner decide tool invocation sequences — is it rule-guided or purely prompt-driven?

2- Have you tested cross-domain transfer (e.g., from OBS to other ancient scripts or multimodal corpora)?

3- How sensitive is OracleAgent’s performance to the backbone LLM (e.g., DeepSeek-V3.1 vs. GPT-4o)?

4- Could you provide more detail on annotation guidelines for the multimodal knowledge base to ensure reproducibility?

5- In the facsimile generation task, what is the average inference cost, and could this be scaled to millions of rubbings efficiently?

---

### Official Review · Reviewer_yA4k · 2025-10-30

**Soundness:** 2
**Presentation:** 3
**Contribution:** 1
**Rating:** 2
**Confidence:** 3

**Summary:**

The paper introduces an agentic pipeline to expediate Oracle Bone Script Research. They tackle knowledge base generation, tool specifications and leverage task-oriented planning with LLMs. They show significant improvements on OBS tasks over using MLLMs with in-context prompting.

**Strengths:**

1.	OBS Research seems to be a challenging task where researchers navigate multiple large heterogeneous sources of using domain specific tools to perform required analysis. The proposed agentic framework automates this process and shows high accuracy on OBS tasks, significantly expediting research work.
2.	The agentic system is well designed with a focus on knowledge base curation, followed by tool specification and orchestration of these into an agentic system where an LLM makes and executes a plan to solve the user’s query.

**Weaknesses:**

1.	Limited novelty and insights for the ML community as the agentic task is highly specialized to OBS research, is relatively simple where planning needs to only consider 7 tools which have clear purposes, in contrast to desktop/code agentic frameworks where complex reasoning and planning is required using a large number of tools. The saturated benchmark scores also indicate this.
2.	The evaluation pipeline used for MLLMs is unclear. Oracle Agent is evaluated with full access to a domain KB and specialized tools while baseline MLLMs are run closed book, making it hard decouple resource advantage and model/agent ability.
3.	Deeper analysis of failure patterns of the agentic system like incorrect tool calls, recovery patterns is missing.

**Questions:**

1.	Could the authors provide leaderboards in closed-book and tool augmented scenarios?
2.	Could the authors provide an analysis of failure rates for tool calls and recovery patterns?

---

### Official Review · Reviewer_tK1p · 2025-11-01

**Soundness:** 3
**Presentation:** 3
**Contribution:** 2
**Rating:** 2
**Confidence:** 4

**Summary:**

OracleAgent is a domain-specific, tool-using agent for Oracle Bone Script research that couples an LLM “brain” with detection/retrieval/generation tools over a custom multimodal knowledge base, and it allegedly beats general MLLMs, such as GPT-4o, on OBS tasks and speeds expert workflows.

**Strengths:**

- This paper introduce a large, curated domain KB (1.4M crops, 80K texts, 3K docs, dictionary; plus 15K pixel-level links) that, if released, could materially help the field.
- A thoughtful, end-to-end framing of practical OBS workflows with the right primitives (modalities, retrieval targets, facsimiles). The Fig. 2 architecture and Fig. 3 interaction flow make it concrete.
- The case study with precision/recall/coverage is stronger than a purely qualitative vignette.

**Weaknesses:**

- Unfair: Most baselines are general-purpose MLLMs without access to the same domain KB; OracleAgent’s advantage could primarily stem from the private data + tools rather than the agent design. The paper doesn’t ablate “with/without KB/tools” vs. the same LLM. (No ablation reported.)
- “OracleAgent-Bench” extends OBI-Bench, but details on train test separation vs. their massive KB are thin; risk of overlap or leakage isn’t audited. (The paper lists sources and sizes, not leakage checks.)
- The “Brain” is described abstractly (state st, utility R, plan πt), but there’s no concrete algorithm, planner evaluation, or latency/cost profile. It reads like a classic tool-call pipeline more than an evaluated planner.
- Code/data to be released “after acceptance”; without them, many results (especially the case study retrieval) aren’t verifiable. Maybe at least release a subset of data and code.
- ID/KID/LPIPS/SSIM are generic; for epigraphy, stroke-level fidelity, legibility, and editor usability matter. The paper lacks a human study with epigraphers for generated facsimiles.
- No report of expert time-on-task reductions beyond a single case; licensing/rights for some corpora and rubbings are not discussed in depth. (Ethics statement is generic.)

**Questions:**

See weakness

---

### Note · Authors · 2025-11-13

I have read and agree with the venue's withdrawal policy on behalf of myself and my co-authors.